# Systematic review of lay consultation in symptoms and illness experiences in informal urban settlements of low-income and middle-income countries

Chinwe Onuegbu  , Maxwell Larweh, Jenny Harlock, Frances Griffiths 

Division of Health Sciences, Warwick Medical School, University of Warwick, Coventry, UK

**Correspondence to**
Chinwe Onuegbu;
chinwe.onuegbu@warwick.ac.uk

## ABSTRACT

**Objectives** Lay consultation is the process of discussing a symptom or an illness with lay social network members. This can have positive or negative consequences on health-seeking behaviours. Understanding how consultation with lay social networks works in informal urban settlements of low-income and middle-income countries (LMICs) is important to enable health and policy-makers to maximise its potential to aid healthcare delivery and minimise its negative impacts. This study explored the composition, content and consequences of lay consultation in informal urban settlements of LMICs.

**Design** *Mixed-method* systematic review.

**Data sources** Six key public health and social science databases, Google Scholar and reference lists of included studies were searched for potential articles.

**Eligibility criteria** Papers that described discussions with lay informal social network members during symptoms or illness experiences.

**Data analysis and synthesis** Quality assessment was done using the Mixed Methods Appraisal Tool. Data were analysed and synthesised using a stepwise thematic synthesis approach involving two steps: identifying themes within individual studies and synthesising themes across studies.

**Results** 13 studies were included in the synthesis. Across the studies, three main categories of networks consulted during illness: kin, non-kin associates and significant community groups. Of these, kin networks were the most commonly consulted. The content of lay consultations were: asking for suggestions, negotiating care-seeking decisions, seeking resources and non-disclosure due to personal or social reasons. Lay consultations positively and negatively impacted access to formal healthcare and adherence to medical advice.

**Conclusion** Lay consultation is mainly sought from social networks in immediate environments in informal urban settlements of LMICs. Policy-makers and practitioners need to utilise these networks as mediators of healthcare-seeking behaviours.

**PROSPERO registration number** CRD42020205196.

## Strengths and limitations of this study

► This is the first study to synthesise evidence on the composition, content and consequences of lay consultation in informal urban settlements of low-income and middle-income countries.
► The review adopted a mixed-method design and synthesised evidence from studies with various study designs using a thematic synthesis approach.
► This review's main limitation is that only studies on lay consultation during illness were included; this excludes studies on other healthcare aspects such as prevention and health maintenance.

friends. Informal social networks are an individual's personal ties outside the formal medical system.[1] Lay consultation theory was introduced by Freidson (1970), and its proponents use a combination of function-alist and interactionist perspectives to explain that informal social networks tend to act as 'lay consultants' to persons experiencing illness or health concerns.[2] In acting as lay consultants, the networks might contribute to the process of sense-making for a health situation and offer various forms of social support including, information (eg, lay advice), appraisal (eg, lay evaluation of symptoms), instrumental support (eg, financial assistance) or emotional support (eg, showing sympathy).[3]

Lay consultation differs from other informal health interactions, as it occurs in the context of a symptom or illness experience.[3] Such experiences are often characterised by a heightened need for social support and care, especially when severe and surpass individuals' coping capacity.[4] Lay consultants contribute to health-seeking decisions, and their roles are particularly important in contexts where healthcare access is poor.[5] Some studies have found that lay consultants promote positive health-seeking behaviours,

## BACKGROUND

Lay consultation refers to the process of discussing a symptom or an illness with informal social networks, such as family and

including encouraging prompt formal care-seeking and providing referrals to care providers.[6 7] However, other studies have found that they discourage healthcare-seeking through rumours or contribute to health-seeking delay due to time used in the consultation.[8 9]

This review considers the use and impacts of lay consultation in informal urban settlements of low-income andmiddle-income countries (LMICs). We acknowledge that 'Informal urban settlements' is used interchangeably with 'slums' in the literature, but we adopt the former as the latter may be defamatory.[10] Informal urban settlements are make-shift low resource settings in cities of LMICs, which houses more than 60% of the urban population.[11] These settlements are densely populated and lack basic social and physical amenities (including clean water, proper housing, hygienic environment, security), contributing to a high burden of diseases.[12] Previous studies have found difficulties in accessing comprehensive formal medical care in informal urban settlements,[12 13] and many rely on informal social support to cope with illnesses.[14] Understanding the use and roles of informal social networks during illness in such contexts is therefore important.

Lay consultation may be different in informal settlements for four major reasons. First, informal social networks are the main sources of support during illness.[14] Second, while the demands are high, access to lay consultants can be challenging as the informal networks are limited and transitory.[15 16] Third, creation and sustenance of community social capitals (actual or potential resources obtainable from social networks)[17] are difficult, as informal settlers tend to be detached from non-kin community networks, and are often unable to provide continuous support to others to protect their mental and physical health.[18] Fourth, informal settlers tend to lack access to weak ties, which are the social connections beyond an individual's immediate environment that can facilitate access to wider resources.[19] Thus, given these distinct challenges, it is important to understand the access to, use and consequences of lay consultation in the settlements.

There is an increasing call to engage social networks in facilitating health interventions in informal settlements of LMICs,[20] but this is hinged on understanding how the networks work. This review synthesises available evidence on the use and influence of lay consultants in the settlements to help policy-makers and providers draw on their benefits to maximise value from healthcare and mitigate their negative effects. We aimed to answer three questions: (a) which informal social networks provide lay consultation? (b) what is the content of the lay consultation? (c) what are the consequences of lay consultation on health-seeking behaviours?

## METHODS

A mixed-methods systematic review design was adopted.[21] The design allows for including qualitative, quantitative and mixed-methods empirical studies in a single review. Preferred Reporting Items for Systematic Reviews and Meta-Analyses guidelines were followed in reporting this review.

### Eligibility criteria

We included studies that (a) reported discussions with informal social network members regarding a symptom and illness experience; (b) collected primary data using quantitative, qualitative or mixed methods. The discussions could be initiated by the person with symptoms or illness conditions, a caregiver or someone within their informal social network. The network members had to be lay people, for example, family and friends. We also included studies where people consulted community health workers (CHWs) for advice only and not treatment. CHWs are lay community members, who receive minimal health training to support formal healthcare delivery in hard-to-reach areas in LMICs.[22]

We excluded (a) studies on prevention and general health behaviours, as these are different from symptom and illness situations, which implies the perceived presence of abnormal health condition and need for care.[23] (b) Studies where the focus was on the management of a long-term condition such as HIV or diabetes which had become part of the routine daily life for the individual and there were no new symptoms or illness experiences; (c) lay consultation with informal healers such as traditional medicine practitioners or spiritualists, as they have indigenous medical knowledge and are alternative health providers in LMICs[24] and (d) lay consultation in informal settlements in high-income countries.

### Search strategy

We searched for relevant articles using six electronic databases (Medline, CINAHL, PsycINFO, Web of Science, Scopus and Applied social science index and abstracts) up to September 2020. In each database, we combined keywords and MESH terms associated with *social networks, lay consultation, illness behaviours and informal settlements* and a list of LMICs[25] (see online supplemental file 1), using Boolean and connectors to search for articles. There were no language, document type or time barriers restrictions. Supplementary searches were conducted using Google Scholar and reference lists of retrieved articles.

### Study selection and appraisal

We collated the identified records into Endnote for deduplication.[26] The deduplicated articles were then transferred into Rayyan, a web-tool for systematic reviews that allows collaborators to independently screen title and abstracts and compare their decisions,[27] for further screening. Two researchers conducted the screening. After an initial title and abstract screening, a full-text appraisal of the identified articles was carried out against the study inclusion criteria. At each stage, the two researchers screened the papers independently and then

met to compare their decisions. Any disagreements were resolved by consulting a third reviewer.

## Data extraction

A data extraction form was developed to extract the characteristics and findings of the included studies. A pilot testing involving data extraction from three articles was done by the primary reviewer and reviewed by two other researchers. Data extracted were publication details, country, study area, study design, objectives, sampling, data collection, analysis techniques and relevant findings from each study. The remaining data extraction was done by the primary reviewer and checked by a second and third reviewer.

## Quality assessment

The Mixed Method Appraisal Tool (MMAT),[28] which contains criteria for assessing multiple study designs, was used to assess the methodological quality of the included papers. The MMAT assess each study design using five criteria, and a score of 1 is given when a publication meets each criterion. Each study's overall quality score ranged from 0/5 (no criterion is met) to 5/5 (all criteria is met). We described the quality assessment scores of included studies but did not exclude any study based on the scores; this was to ensure that we gathered data from all studies that have been done on the research topic. However, where it was important for understanding study results, we highlighted the methodological limitation of a study when presenting its results. Two researchers independently assessed the papers' quality, and all disagreements were discussed with a third researcher.

## Data analysis and qualitative synthesis

We initially summarised the characteristics of all included studies. We then used a stepwise thematic approach to synthesise the qualitative, quantitative and mixed-methods findings from the included studies.[29] This approach consisted of two stages: (i) identification of patterns and themes within each study based on the research questions, and (ii) synthesising of themes across studies. In the first stage, we used the research questions to provide an overarching structure for the analysis. Three researchers independently read each study's extracted findings and identified codes under each research question. We then compared the identified codes and generated themes under each research question. In the second stage of synthesis, we compared the generated themes for each research question across studies and developed a final list of analytical themes.

## RESULTS

### Overview of included studies

The systematic search strategy yielded 5740 records. Following deduplication and title and abstract screening, 86 articles were identified for full-text screening. Of these, 13 articles met the inclusion criteria and were included

for qualitative thematic synthesis. The information on the screening and selection process is shown in figure 1.

The 13 included studies were conducted in five countries: six studies in India, three in Kenya, two in Bangladesh, one in Nicaragua, and one in the Philippines. The timing of publication ranged from 1998 to 2019. The studies focused on different care-seeking behaviours and reported findings beyond lay consultation; thus, only findings relevant to our review were included. The data presented on lay consultation in almost all papers except[30] were relatively little, and the phenomenon was not discussed in great detail. Relevant qualitative data were reported in 10 papers, and quantitative data were reported in three papers. Information on the study characteristics is shown in table 1.

Using the MMAT to assess the methodological quality of the included papers, we found almost all studies that reported qualitative data met all the criteria, except one study[31] that scored 4/5 due to insufficient data to support the results presented. Of the three quantitative descriptive studies, only one[32] met all the criteria. The other two papers[33 34] scored 2/5 and 3/5, respectively, as information on the sample representativeness, low response rates and analysis procedures were not adequately provided. The detailed quality assessment of the included studies is present in table 2.

In the following sections, we present the thematic analysis and synthesis of findings based on our three research questions. The research questions were used as overarching themes, and descriptive themes were identified and discussed under each one. An overview of the identified themes and their associated references is presented in the concept matrix in table 3.

### Who provides lay consultation?

We identified three main categories of lay consultants: kin networks, networks by association and significant others in the community.

### Kin networks

Kin networks comprising family members and relatives were reported as lay consultants in all but one[33] of the included studies. Specifically, spouses, parents, in-laws and grandmothers were mentioned. Nuclear family members were particularly identified as critical sources of support and advice during illness. One study reported that nuclear family members were usually the first point of contact during illness and were more trusted with information about stigmatising illnesses.[30]

### Non-kin associates

These refer to persons connected to an individual through friendship ties, engagement in similar activities or physical proximity. We found three groups under this category: friends,[34 35] neighbours[36–39] and colleagues.[30 35] Neighbours in shared structures were sometimes consulted or volunteered health advice when they noticed unusual symptoms in others.[38] While colleagues were common

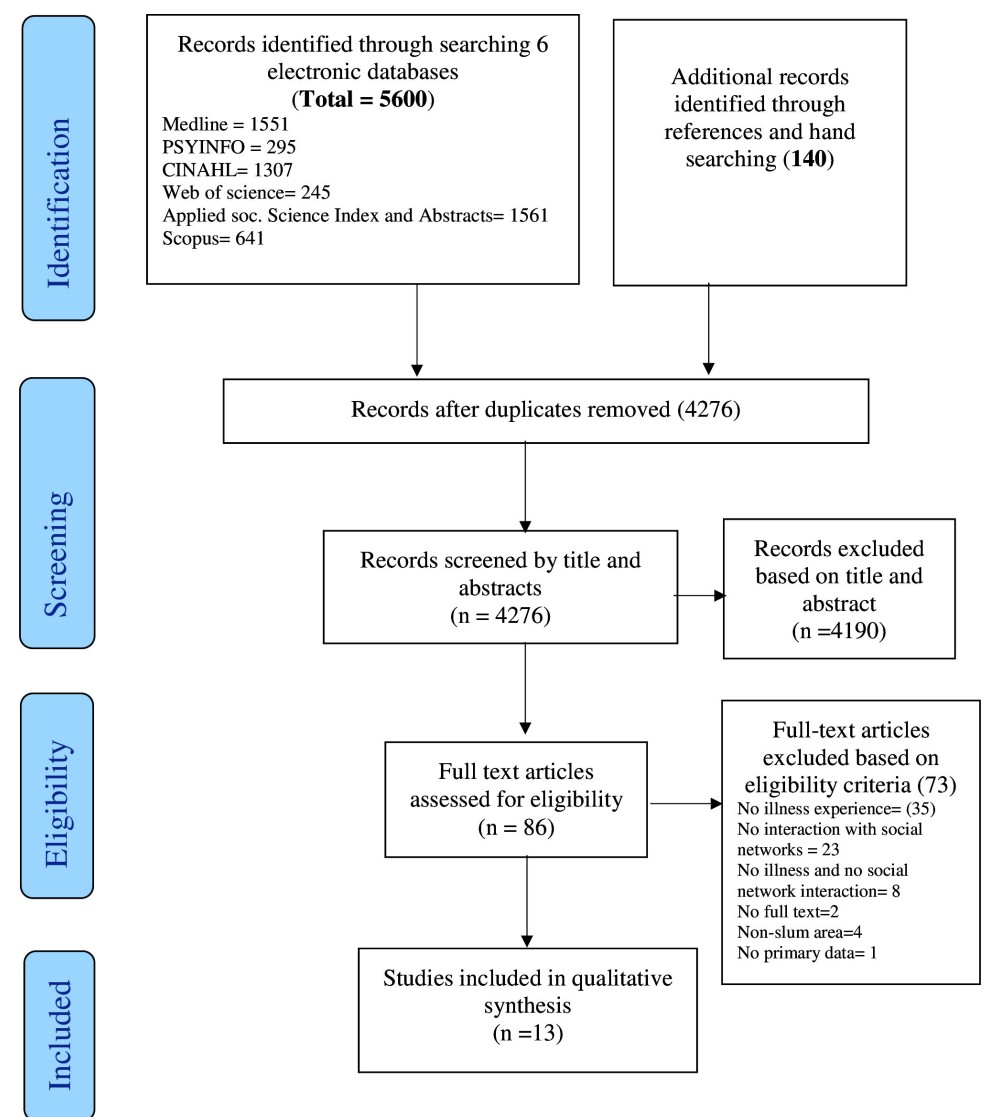

**Figure 1** Summary of the literature search process following PRISMA reporting guidelines. PRISMA, Preferred Reporting Items for Systematic Reviews and Meta-Analyses.

sources of advice, employers discouraged discussing health problems in the workplace.[30] As informal settlers were mostly casual workers, extra caution was employed in seeking health advice from others to avoid wasting time or showing signs of poor health to their employers.[30]

### Significant others in the community
These were influential community members consulted for specific kinds of advice during illness. CHWs were consulted for information about formal care services.[33 38] Village heads and opinion leaders are consulted when there was difficulty in making health-seeking decisions within the home or during health emergencies.[40] Persons with similar illnesses served as mentors and sources of encouragement to patients.[30]

### What is the content of the lay consultation?
We identified four themes that describe the content of social interactions with social networks during illness. They include: Asking for suggestions and opinions

about managing symptoms/illnesses, making and negotiating care-seeking decisions with household decision-makers, seeking material and non-material forms of support and non-disclosure due to personal or social circumstances. These themes denote how people participate or do not participate in lay consultation based on their health needs, personal characteristics and social circumstances.

### Asking/receiving suggestions and opinions about how to manage symptoms/illnesses
In most studies, lay consultation was used to seek or receive suggestions and opinions about managing symptoms or illnesses.[30–39 41] This kind of conversation involves an obvious search for, or offer of, information at different points within illness pathways. The kinds of advice sought were about symptom evaluation, medications, herbal treatments and healthcare facilities.[30–37 39 41]

**Table 1** Study characteristics

| Paper | Country | Study area | Main objective/research question of the study | Study design relevant to the review | Method of data collection and analysis relevant to the review | Participants relevant to the review |
|---|---|---|---|---|---|---|
| Ailinger et al[41] 2004 | Nicaragua | A Barrio in Managua | To examine the use of herbal remedies in treating common illnesses. | Qualitative | Open-ended interviews; content analysis | 25 women and 2 men who had been ill in the 3 months preceding the survey. |
| Amuyunzu-Nyamongo and Nyamongo[37] 2006 | Kenya | Kawangware, Korogocho, Viwandani and Njiru, in Nairobi | What actions do mothers take during childhood illnesses? | Qualitative | In-depth interviews; thematic analysis | 62 mothers of under-5 children who had been sick in the previous 3 months preceding the study. |
| Angeli et al[39] 2018 | India | Slum areas of Ahmedabad, Gujarat | To understand the drivers of choice among Bottom of the Pyramid (BOP) patients by employing grounded theory methods; to find out BOP patients' main concerns and social processes in making consumption choices. | Qualitative | Interview schedules/ content analysis | 21 slum residents |
| Bhandari et al[31] 2002 | India | Two slums in Delhi | To obtain insights into the processes underlying infant deaths to help identify preventive interventions which may bring down infant mortality rates further. | Verbal autopsies | Semi-structured questionnaire developed and validated by the WHO | Not clear |
| Das et al[30] 2018 | India | Four urban slums in Kolkota and Banglore | To uncover the many facets of lay decision-making before future action is taken and the reasons underpinning illness-expressing behaviour among Indian urban slum dwellers. | Qualitative | Semi-structured in-depth interviews; thematic analysis | 218 participants (105 men and 113 women) that experienced illnesses in 1 year preceding the study. |
| de Zoysa et al[36] 1998 | India | A dense slum settlement in New Delhi | To assess maternal recognition and interpretation of illness in young infants, and identify constraints to the adequate provision of care for the illness. | Focused ethnographic study | In-depth interviews using narratives; focused ethnography approach to analysis | 37 mothers of young infants (between 1 week and 2 months of age) who complained of a recent or current illness in their young infants. |
| Essendi et al[40] 2011 | Kenya | Viwandani and Korogocho, Nairobi | To investigate poor urban Kenyan men and women's views on the factors that hinder the uptake of formal obstetric care services. | Qualitative | Focus group discussions; thematic analysis | 16 focus groups; Women who had life-threatening obstetric complications and failed to seek healthcare in the 2 years preceding the survey; their partners, opinion leaders, Traditional Birth Attendants and older women in the communities. |

Continued

**Table 1**   Continued

| Paper | Country | Study area | Main objective/research question of the study | Study design relevant to the review | Method of data collection and analysis relevant to the review | Participants relevant to the review |
|---|---|---|---|---|---|---|
| Ghosh et al[34] 2010 | India | Patpur slum, Bankura, West Bengal | To determine the prevalence of chest symptomatics among the study population, study their healthcare seeking behaviour and identify the underlying sociodemographic correlates | Cross-sectional, descriptive community-based study | Semi-structured questionnaire; proportions | 64 people with cough for 3 weeks or more with or without haemoptysis, fever, chest pain, weight loss and/or night sweating. |
| Hu et al[38] 2012 | The Philippines | Payatas slum in Quezon city. | To characterise tuberculosis care-seeking in Payatas and identify facilitators and barriers at the individual, household, community and health-system levels from the perspective of the community. | Qualitative study design using multimethods | Semi-structured in-depth interviews and focus group discussions; thematic analysis | 13 female patients receiving treatments for tuberculosis from health centres |
| Taffa et al[32] 2005 | Kenya | Kawangware, Korogocho, Njiru and Viwandani | To assess the healthcare utilisation among slum residents in Nairobi City, Kenya. | A pilot study by the Nairobi Urban Demographic Surveillance System (NUDSS). | Child morbidity interview questionnaires adopted from UNICEF multiple indicator cluster survey (MICS2) and the WHO integrated management of childhood illness/percentages. | 264 children in whom morbidity was reported at least once during a 9-month observation period. |
| Uzma et al[42] 1999 | Bangladesh | Slums in the four wards of Motijheel thana, Dhaka city | To describe the circumstances of women following childbirth by exploring patterns of birth-related illnesses, their healthcare seeking behaviour and their beliefs and attitudes relating to both their illnesses and any services they have received. | Qualitative | Semi-structured interviews and focus group discussion; not reported | 122 women with recent childbirth experience; 8 women in the community below 50 years of age. |
| van der Heijden et al[35] 2019 | Bangladesh | Kamrangirchar | To document how people perceive their health and care options and seek healthcare within the community. | Qualitative descriptive explanatory design | In-depth interviews, using a flexible participant-led approach based on a topic guide; inductive thematic analysis | 13 women; 14 factory workers |
| Waghela et al[33] 2018 | India | Slums of Durg and Bhilai | To understand the role of Mitanins (community health workers) in health seeking of their slum population. | Descriptive cross-sectional study | Prestructured questionnaire/descriptive statistical analysis | 500 slum residents |

## Making and negotiating care-seeking decisions with household decision-makers

Two studies reported that people talked about their illness with household decision-makers in control of decisions about the use of healthcare facilities.[35 42] Both studies found that this was common among women with limited autonomy and financial power, who required their husbands' approval and support. In contrast, more financially independent women and those living separately from their husbands reported more autonomy in healthcare-seeking decisions.[35]

## Seeking material and non-material forms of support

Some people discussed their health conditions with others in their networks to seek material and non-material forms of support.[30 36 38] The forms of support were cash loans or child care support to enable access to formal healthcare services, reassurance and emotional support to cope with illness and companionship to healthcare facilities.[30 36 38]

## Non-disclosure due to personal or social circumstances

Four studies discussed how people avoided consulting social network members for advice or support due to the nature of the illness, sociocultural factors and perceived lack of community support.[30 35 38 40] In two studies, participants reported that they avoided discussing stigmatising illnesses with neighbours to avoid being gossiped about, isolated or discriminated.[38 40]

Stereotyped conditions, particularly sexual and reproductive health issues, were also less discussed with social

**Table 2** Quality assessment using mixed-methods appraisal tool

| Studies | Study design | Methodological quality criteria | | | | | Overall score |
|---|---|---|---|---|---|---|---|
| | Qualitative | Is the qualitative approach appropriate to answer the research question? | Are the qualitative data collection methods adequate to address the research question? | Are the findings adequately derived from the data? | Is the interpretation of results sufficiently substantiated by data? | Is there coherence between qualitative data sources, collection, analysis and interpretation? | |
| Allinger and Zamora[41] 2004 | | Yes | Yes | Yes | Yes | Yes | 5 |
| Amuyunzu-Nyamongo and Nyamongo[37] 2006 | | Yes | Yes | Yes | Yes | Yes | 5 |
| Angeli et al[39] 2018 | | Yes | Yes | Yes | Yes | Yes | 5 |
| Bhandari et al[31] 2002 | | Yes | Yes | Yes | No | Yes | 4 |
| Das et al[30] 2018 | | Yes | Yes | Yes | Yes | Yes | 5 |
| Essendi et al[40] 2010 | | Yes | Yes | Yes | No | Yes | 5 |
| Heijden et al[35] 2019 | | Yes | Yes | Yes | Yes | Yes | 5 |
| Hu et al[38] 2012 | | Yes | Yes | Yes | Yes | Yes | 5 |
| de Zoysa et al[36] 1998 | | Yes | Yes | Yes | Yes | Yes | 5 |
| Uzma et al[42] 1999 | | Yes | Yes | Yes | Yes | Yes | 5 |
| | Quantitative (descriptive) | Is the sampling strategy relevant to address the research question? | Is the sample representative of the target population? | Are the measurements appropriate? | Is the risk of non-response bias low? | Is the statistical analysis appropriate to answer the research question? | |
| Waghela et al[33] 2018 | | Yes | Cannot tell | Yes | Cannot tell | Cannot tell | 2 |
| Taffa et al[32] 2005 | | Yes | Yes | Yes | Yes | Yes | 5 |
| Ghosh[34] 2010 | | Yes | Cannot tell | Yes | Cannot tell | Yes | 3 |

**Table 3**  Concept matrix identifying main themes

| Studies | Themes | | | | | | | | | | |
| --- | --- | --- | --- | --- | --- | --- | --- | --- | --- | --- | --- |
| | Who provides lay consultation? | | | What is the content of lay consultation? | | | | What are the consequences of lay consultation on health-seeking behaviours? | | | |
| | | | | | | | | Positive | | Negative | |
| | Kin | Non-kin associates | Significant others | Asking/ receiving suggestions or opinions | Making and negotiating health-seeking decisions | Seeking material and non-material forms of support | Non-participation due to personal or social circumstances | Motivate formal care-seeking | Influence positive attitudes towards formal health providers | Non-compliance with medical advice due to contrary suggestions from social network members | Poorly-communicated advice from social network members contribute to poor health-seeking behaviours |
| Ailinger et al[41] 2004 | * | | | * | | | | | | | |
| Amuyunzu-Nyamongo and Nyamongo[37] 2006 | * | * | | * | * | | | | | | |
| Angeli et al[39] 2018 | * | * | | * | | | | | * | | |
| Bhandari et al[31] 2002 | * | | | * | | | | | | * | |
| Das et al[30] 2018 | * | * | * | * | | * | * | | | | |
| de Zoysa et al[36] 1998 | * | * | | * | | * | | | | | |
| Essendi et al[40] 2011 | * | | * | | * | | | | | | |
| Ghosh et al[34] 2010 | * | * | | * | | | | * | | | |
| Hu et al[38] 2012 | * | * | * | * | | * | * | | | | |
| Taffa et al[32] 2005 | * | | | * | | | | | | | |
| Uzma et al[42] 1999 | * | | | | * | | * | | | | * |
| van der Heijden et al[35] 2019 | * | * | | * | * | | | | | | |
| Waghela et al[33] 2018 | | | * | * | | | | * | | | |

networks due to shyness,[35] or because it was culturally inappropriate to do so, especially with the opposite sex.[30]

Gender norms also influenced non-disclosure to social networks.[30] A qualitative study exploring the illness disclosure patterns among a small sample of men and women in an Indian settlement found that men were less likely than women to seek health advice from their networks.[30] While the men expressed that seeking advice was feminine, the women reported that they were expected to report health concerns to others and seek advice. Consequently, men reportedly had fewer health discussion and social support networks than women. However, while this study points out differences in the composition and size of men and women's networks, this was a qualitative study done among a small sample and there were no statistical tests of differences in men and women's lay consultation networks.

Perceived lack of community social support and interdependence discouraged some people from disclosing their conditions to others to seek their advice or support.[30] One study reported that some people did not seek support from others during illness because they believed that community members faced several challenges that weakened their capacity to help others.[30] For instance, a participant in the study explained: 'Everyone here is busy fixing their problems, so it is as though I neither have the patience to discuss symptoms nor do people here have the patience to listen'. Thus, the perceived lack of reciprocity and support in the community influenced people to manage illness episodes independently or seek support only from their household members.

### What are the consequences of the lay consultation on health-seeking behaviours?

We identified four themes on the consequences of lay consultation on health-seeking behaviours. Two were on the positive consequences ('motivate formal care-seeking', 'influence positive attitudes towards formal health providers') and two on the negative consequences ('non-compliance with medical advice due to contrary suggestions from social network members, 'poorly-communicated advice from social network members contribute to poor health-seeking behaviours).

#### Positive consequences
*Motivate formal care-seeking*
Two studies discussed how lay consultants influenced people to seek formal care.[33 34] In one of the studies, 64 persons with chest symptoms in an Indian informal settlement were surveyed to understand their healthcare-seeking behaviour.[34] The study found that about 44% of the respondents visited formal healthcare facilities based on advice from their family and friends. Similarly, another study found that advice CHWs influenced formal care-seeking for chronic or acute illnesses in an informal settlement in India.[33]

*Influence positive attitudes towards formal health providers*
Social networks' influence in encouraging positive attitudes towards formal health providers was discussed in one study.[39] The study explored the decision-making behaviours of bottom-of-the pyramid patients (individuals with comparatively weaker earnings) and found that some patients chose and committed to healthcare providers recommended by their family members and neighbours as trustworthy. This was particularly useful as there was a general attitude of mistrust and uncertainties about formal healthcare services.

#### Negative consequences
*Non-compliance with medical advice due to contrary suggestions from social network members*
This was reported in a study that explored the pathways to infant mortality among parents of deceased infants in a settlement in India.[31] Among other reasons, advice from family members encouraged non-compliance with the prescribed medical advice on hospitalisation, contributing to infant mortalities. However, this study did not provide enough information on the family members' advice to understand the negative impacts further.

*Poorly communicated advice from social network members contribute to poor health-seeking behaviours*
One study reported that intrusive and embarrassing comments from informal social networks deterred people with tuberculosis (TB) symptoms from seeking appropriate care.[38] Some people felt ashamed when neighbours commented on their TB symptoms and refused to disclose their TB status or obtain formal medical diagnoses.

### DISCUSSION AND CONCLUSION
This review found that illnesses were managed with advice and support from lay social networks in informal settlements of LMICs. Closely-knit networks and others in the immediate environment dominated lay consultation, suggesting a lack of extensive lay consultation networks. Lay consultation was mainly used to obtain information and practical resources to aid healthcare-seeking. However, factors such as perceived illness taboos, gender norms or perceived lack of community support discouraged some people from seeking advice or support from others. The review also showed how lay consultants' advice contributed positively and negatively to health-seeking behaviours.

The finding that nuclear family members dominated lay consultation raises two major issues. First, it reaffirms the importance of the family as a critical source of health-related social support in informal settlements of LMICs and suggests that people who get cut-off from their families due to forceful eviction by city authorities may be exposed to increased vulnerabilities stemming from loss of primary means of social support, as highlighted by other studies on displacement.[43–45]

Second, it suggests that the informal settlers may have lacked access to more extensive networks beyond their immediate environment. Such networks, known as weak ties, are important for linking individuals to wider sources of support.[46] On the contrary, reliance on strong ties (persons who share similar characteristics and are strongly bonded to each other such as family members) limits individuals to resources within the network.[46] Networks of weak ties, for example, friends from religious or economic groups are hard to create or maintain in transient urban communities as they require commitments, however, they provide potential benefits.[47]

Furthermore, the review showed that lay consultation was beyond the functions of lay consultants in illness pathways as emphasised by lay consultation theorists, but it was also a tool for control and discrimination. Women could not make health-seeking decisions without consulting their husbands because they were considered to have lesser financial and decision-making power. This relates to a broader problem of 'feminisation of poverty' in low-income environments, which implies that women in low resource contexts are overburdened with production and reproduction activities but occupy weaker power positions in the household.[48 49]

In addition, some of the included studies reported that network members used lay consultations to stigmatise and discriminate against persons with negatively labelled conditions. Other studies have noted that gossiping in informal urban settlements is common,[50 51] and it is used as a tool for punishing social deviants.[52]

### Strengths and limitations

By focusing specifically on lay consultations for symptom or illness experiences, our analysis gave insights on its use, multidimensional functions and health-related consequences in informal settlements of LMICs. However, we identified only a small number of papers directly relevant to our research questions, and within these papers, lay consultation was often not the main focus. We excluded studies on consultations with social network members to obtain support for prevention, health maintenance and chronic illnesses, so our review's findings may not be exhaustive of people's health-related social support networks. Our review identified some reasons people did not access advice or support during illness from their social network members. However, we did not set out to find this evidence; a different approach to searching may reveal more literature on this topic.

### Implications for policy, practice and research

Lay consultation networks should be recognised as potential mediators for healthcare seeking in policy-making for informal settlements of LMICs. Accessible and credible health information resources are important for individuals to obtain formal advice about their conditions and to check the information obtained from their social networks. Healthcare practitioners need to recognise that many conversations about illness and symptoms occur with lay social networks. Asking about this can help practitioners understand why a patient seeks their help and their expectations.

### CONCLUSION

Lay consultation is an integral process in the pathways to healthcare in informal settlements of LMICs. Yet, there are relatively few studies that have examined its use and impacts on health-seeking behaviours. Future research is needed to understand the structural characteristics of lay consultation networks (such as network size, density, composition) in informal settlements. This would provide insights into how lay social networks that provide health-related support are structured and utilised.

**Acknowledgements** Chinwe Onuegbu's PhD is sponsored by the Chancellors International Scholarship, University of Warwick, UK.

**Contributors** CO conducted the search, screening, quality assessment and wrote the entire review. CO is reponsible for the overall content as guarantor. ML conducted the second screening of the articles and provided input on the data collection process. JH and FG were involved in the design of the research and research protocol, provided input at all stages of the review, reviewed and revised the papers. All authors approved the final version of the paper.

**Funding** This research was funded by the National Institute for Health Research (NIHR) Global Health Research Unit on Improving Health in Slums using UK aid from the UK Government to support global health research. The views expressed in this publication are those of the author(s) and not necessarily those of the NIHR or the UK Department of Health and Social Care.Grant reference number 16/136/87.

**Competing interests** None declared.

**Patient consent for publication** Not required.

**Ethics approval** Since this was a systematic review of publicly available data, and no primary data were collected, ethics approval was not applicable.

**Provenance and peer review** Not commissioned; externally peer reviewed.

**Data availability statement** All data relevant to the study are included in the article or uploaded as supplementary information.

**ORCID iDs**
Chinwe Onuegbu http://orcid.org/0000-0001-6372-9390
Frances Griffiths http://orcid.org/0000-0002-4173-1438

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
