## [Reviewer comments · BMJ Open]

ARTICLE DETAILS

TITLE (PROVISIONAL)	A systematic review of lay consultation about symptoms and illness experiences in informal urban settlements of Low- and Middle-Income Countries
AUTHORS	Onuegbu, Chinwe; Larweh, Maxwell; Harlock, Jenny; Griffiths, France

VERSION 1 – REVIEW

REVIEWER	Montgomery, Shannon Florida State University
REVIEW RETURNED	02-Jun-2021

GENERAL COMMENTS	This is an interesting mixed methods systematic review that aims to understand the involvement of lay social networks in illness and symptom management in low- and middle- income countries (LMICs). There are several major concerns to this study outlined below. In particular, there is a very narrow focus on health. Specifically, only illness and symptoms have been included. Why have mental health and health behaviors not been included? Furthermore, there is a loose connection to LMICs and little mention of informal urban settlements. Further concerns are outlined below: Title 1. The title does not suggest any connection with health. I would recommend that you re-consider the wording of the title to include health, healthcare or illness terminology. Background 2. The main concern with the background is that there is a weak argument supporting the need for this study. The background contains no mention of a strong theoretical argument supporting the research aims.3. Page 3, line 19: Please provide a definition of social networks. Many readers will be unfamiliar with the term 'social networks' or may assume that they are online social networks. Furthermore, throughout the background the social networks definition was poor.4. Page 3, line 27-53: I am a little confused as to why the example provided relates only to a high-income country (UK), when the focus is on LMICs?
--

5. There is little emphasis on specific networks in LMICs.

Methods

1. Page 5, line 17: One of your research questions is 'who provides lay consultation', so why have you outlined in the eligibility criteria that social network members had to be related to the individual, such as family or friends, or community health workers? Surely this will narrow your findings?
2. Page 5, line 26: why were studies that looked at other health-related purposes besides symptoms or illness management included?
3. Page 7, line 11: patient and public involvement declaration should be moved to the end.

Results

1. Page 12: You state that the themes were based on the research questions. Where is the rationale to support these research questions?
2. Page 12, line 5: For the third theme 'what are the consequences of lay consultation on health-seeking behaviors' – limiting the search to only symptoms and illness experience narrows your findings significantly.
3. Why have you excluded other health related purposes'? surely you have excluded other important findings relating to other health-related outcomes and health behaviors, in particular. For example, in LMICs, they have increased prevalence of certain health behaviors, i.e. smoking, so why were they excluded?
4. Page 13, line 12: this paragraph is missing citations.

Discussion

1. Page 17, line 29: How did you assess if other formal forms of treatment/ health-care were also provided in conjunction with lay consultations?
2. Page 17, line 30-32: it is unsurprising that there was little by way of lay consultation as you state that you termed social networks to be related individuals in your eligibility criteria.
3. Page 19, line 12: You introduce new terms in this paragraph but you have not explained them, i.e. 'weak ties' have not been defined.
4. Page 12, line 14: please define 'norms of reciprocity'.

REVIEWER	Rueger, Jasmina Wageningen University, Business Management & Organisation
REVIEW RETURNED	24-Jun-2021

GENERAL COMMENTS	Overall, the manuscript is a potentially interesting contribution to the literature on lay consultation and its effects on treatment decision-making in LMICs. Especially the implications for policymakers and HCPs in LMICs, if further developed, sound promising. The study focuses on a relevant gap, was conducted in a methodologically sound manner and has produced some interesting results, however, I believe that some of these points have not been developed fully yet. Please see below for a breakdown of these issues per point on the evaluation form. As such, some of my points were repeated if they were relevant for multiple aspects. In terms of setup (Point 1 of the evaluation form), I believe the authors have described the context (LMICs) and the effect it has on the network as found in previous literature well. Further, the authors have provided interesting background information about how social network members may encourage or dissuade others from seeking treatment in that context. However, in its present form, the manuscript seems to mostly rely on arguments in favour of the dissuasion/delays, not potentially positive effects (e.g., p.4, l. 43-53). Given that, in the end, the authors identify an even number of positive and negative consequences, it would be helpful to give examples that show “both sides” early on already. On a separate note, I believe it would be helpful to add the distinction between different types of support (e.g., emotional, information and network) in the background section as all these concepts are implied in this study but the connection to prior literature was not drawn. I have elaborated on this point further below (Point 8). Point 3 (study design): The exclusion criteria could be explained better (see p.6, l. 26-32). Namely, there could be more reasoning as to why (a) prevention and (b) long-term chronic disease were excluded. (c) and (d) are well-explained. However, I am somewhat confused about why according to (c) “lay consultation with informal healers such as traditional medicine practitioners or spiritualists” was excluded “as they are health providers“ but at the same time studies 33 and 38 consider community health workers. Why weren’t they excluded but instead their samples were deemed to be relevant for the research question of this study? Point 8 (recent and relevant references): Some references are rather outdated and could be supported by newer studies. For instance, on page 4 (lines 26-31) the authors write about informational and emotional support. First, I believe it would be helpful to actually name these concepts as established in the literature (e.g. Smailhodzic, E., Hooijsma, W., Boonstra, A. et al. Social media use in healthcare: A systematic review of effects on patients and on their relationship with healthcare professionals. BMC Health Serv Res 16, 442 (2016). https://doi.org/10.1186/s12913-016-1691-0), and second, doing so would give offer an opportunity to introduce newer studies. Although the study I have suggested focuses on a different context (online communities which also often provide lay advice), it synthesises the different types of social support well. In particular,
---

	“network support” may also be relevant to consider for the scope of the submitted manuscript. Point 9 (research objectives met): This point suffers as there are some findings that weren't explained fully. For instance, later there are considerations about digital communication and other network aspects which were used to answer the research question and address the overall objective of informing policymakers. However, I would be interested to read more specific information about the circumstances, potential interventions, etc. if this is to be used as a base for the policy implications later on. Point 10 (results presented clearly): Overall, the results are well-written. However, p.17 lines 17-19 are unclear due to the sub-sentence (“It was found that family and friends' advice was the main reason why about 44% of the participants chose, and 25% of the participants, formal health facilities.”). Further, the final negative consequence, poorly communicated advice, has not been explained fully. It appears that patients may be less inclined to visit a HCP but this is not completely clear based on the example given. Point 11 (discussion justified by results): For the most part, the discussion is justified. However, in my opinion, there are some new considerations that do not find a good connection to the rest of the study yet. First, the network considerations (e.g., weak ties) come rather suddenly. These are interesting points and as such, should be elaborated on in greater detail in the results and background information. Further, the remarks about digital communication are good. However, they could also be introduced in a more coherent and applied manner. The current description is rather dense and loose from the rest of the study. Nevertheless, it is helpful to read the authors' reflections on the extent to which digital communication could reduce inequalities while also acknowledging the limitations of LMICs. Point 12 (limitations): The limitations are mostly clear. However, I agree that it is quite a limitation that only studies during illness were included. I would be inclined to assume that kin networks also play a substantial role in health promotion and prevention in LMICs and elsewhere. Why were studies excluded that do not explicitly focus on illness? I would like to know more about the specific reasons behind this choice.
--	--

VERSION 1 – AUTHOR RESPONSE

Reviewer 1

Overall comment:

This is an interesting mixed methods systematic review that aims to understand the involvement of lay social networks in illness and symptom management in low- and middle- income countries (LMICs). There are several major concerns to this study outlined below. In particular, there is a very narrow focus on health. Specifically, only illness and symptoms have been included. Why have mental health and health behaviors not been included? Furthermore, there is a loose connection to LMICs and little mention of informal urban settlements. Further concerns are outlined below:

Authors revision:

Thank you for reviewing our manuscript and providing us with feedback.

- We are aware that mental illness and health behaviours such as substance misuse are common in LMICs. We are aware that understanding and responses to these will be shaped by the specific social context and cultural dynamics where people live. However, the specific challenges around mental health conceptualisation and provision within local cultures in LMICs, and related health behaviours such as substance misuse, are beyond the scope of this review. (please see Rathod et al. 2017. Mental health service provision in Low- and Middle-Income Countries

<https://www.ncbi.nlm.nih.gov/pmc/articles/PMC5398308/>).

- We have modified the background to highlight that lay consultation is specific to symptoms and illness experiences on page 3, line 20-28:

Lay consultation differs from other informal health interactions, as it occurs in the context of a symptom or illness experience.(3) Such experiences are often characterised by a heightened need for social support and care, especially when severe and surpass individuals' coping capacity.(4) Lay consultants contribute to health-seeking decisions, and their roles are particularly important in contexts where healthcare access is poor.(5) Some studies have found that lay consultants promote positive health-seeking behaviours, including encouraging prompt formal care-seeking and providing referrals to care providers.(6, 7). However, other studies have found that they discourage healthcare-seeking through rumours or contribute to health-seeking delay due to time used in the consultation.(8, 9)

- We have revised the background and discussion sections and included further references regarding the specific issues/experiences within LMICs and informal urban settlements, as shown below:

Background: This review considers the use and impacts of lay consultation in informal urban settlements of Low-and-Middle-Income Countries (LMICs). We acknowledge that "Informal urban settlements" is used interchangeably with "slums" in the literature, but we adopt the former as the latter may be defamatory.(10) Informal urban settlements are make-shift low resource settings in cities of LMICs, which houses more than 60% of the urban population.(11) These settlements are densely populated and lack basic social and physical amenities (including clean water, proper housing, hygienic environment, security), contributing to a high burden of diseases.(12) Previous studies have found difficulties in accessing comprehensive formal medical care in informal urban settlements,(12, 13) and many rely on informal social support to cope with illnesses.(15) Understanding the use and roles of informal social networks during illness in such contexts is therefore important.

Lay consultation may be different in informal settlements for four major reasons. Firstly, informal social networks are the main sources of support during illness.(14) Secondly, while the demands are high, access to lay consultants can be challenging as the informal networks are limited and transitory.(15, 16) Thirdly, creation and sustenance of community social capitals (actual or potential resources obtainable from social networks)(17) are difficult, as informal settlers tend to be detached from non-kin community networks, and are often unable to provide continuous support to others to protect their mental and physical health.(18) Fourthly, informal settlers tend to lack access to weak ties, which are the social connections beyond an individual's immediate environment that can facilitate access to wider resources.(19) Thus, given these distinct challenges, it is important to understand the access to, use and consequences of lay consultation in the settlements. (page 3, line 29 to page 4, line 23)

Discussion: The finding that nuclear family members dominated lay consultation raises two major issues. Firstly, it reaffirms the importance of the family as a critical source of health-related social support in informal settlements of LMICs and suggests that people who get cut off from their families due to forceful eviction by city authorities may be exposed to increased vulnerabilities stemming from loss of primary means of social support, as highlighted by other studies on displacement.(43-45)

Secondly, it suggests that the informal settlers may have lack access to more extensive networks beyond their immediate environment. Such networks, known as weak ties, are important for linking individuals to wider sources of support.(46) On the contrary, reliance on strong ties (persons that share similar characteristics and are strongly bonded to each other such as family members) limits individuals to resources within the network.(46) Networks of weak ties e.g. friends from religious or economic groups are hard to create or maintain in transient urban communities as they require commitments, however, they provide potential benefits.(47)

Furthermore, the review showed that lay consultation was beyond the functions of lay consultants in illness pathways as emphasised by lay consultation theorists, but it was also a tool for control and discrimination. Women could not make health-seeking decisions without consulting their husbands because they were considered to have lesser financial and decision making power. This relates to a broader problem of “feminisation of poverty” in low-income environments, which implies that women in low resource contexts are overburdened with production and reproduction activities but occupy weaker power positions in the household.(48, 49)

In addition, some of the included studies reported that network members used lay consultations to stigmatise and discriminate against persons with negatively labelled conditions. Other studies have noted that gossiping in informal urban settlements is common,(50, 51) and it is used as a tool for punishing social deviants.(52) (Page 17, line 10 to page 18, line 2)

Title

1. The title does not suggest any connection with health. I would recommend that you reconsider the wording of the title to include health, healthcare, or illness terminology.

Authors revision:

Based on the focus of this review, the original title contains health-related words including symptoms and illness experiences. However, we have made slight adjustments in the title as follows:

Original: A systematic review of lay consultation about symptoms and illness experiences in informal urban settlements of Low- and Middle-Income Countries.

Revised: A systematic review of lay consultation in symptoms and illness experiences in informal urban settlements of Low- and Middle-Income Countries. (page 1, line 1 to 2)

Background

2. The main concern with the background is that there is a weak argument supporting the need for this study. The background contains no mention of a strong theoretical argument supporting the research aims.

Authors revision:

Thank you for highlighting this issue.

- We have provided further information on the theory used on Page 3, line-12-19 as follows:

Lay consultation theory was introduced by Freidson(1970) and uses a combination of functionalist and interactionist perspectives to explain that informal social networks tend to act as “lay consultants”

to persons experiencing illness or health concerns.(2) In acting as lay consultants, the networks might contribute to the process of sense-making for a health situation and offer various forms of social support including, information (e.g. lay advice), appraisal (e.g. lay evaluation of symptoms), instrumental support (e.g. financial assistance) or emotional support (e.g. showing sympathy).(3)

- We have edited the discussion section from page 17 line 22 to page 18 line 2 to clarify the limitations of the theory as revealed by this review:

Furthermore, the review showed that lay consultation was beyond the functions of lay consultants in illness pathways as emphasised by lay consultation theorists, but it was also a tool for control and discrimination. Women could not make health-seeking decisions without consulting their husbands because they were considered to have lesser financial and decision making power. This relates to a broader problem of “feminisation of poverty” in low-income environments, which implies that women in low resource contexts are overburdened with production and reproduction activities but occupy weaker power positions in the household.(48, 49)

In addition, some of the included studies reported that network members used lay consultations to stigmatise and discriminate against persons with negatively labelled conditions. Other studies have noted that gossiping in informal urban settlements is common,(50, 51) and it is used as a tool for punishing social deviants.(52)

3. Page 3, line 19: Please provide a definition of social networks. Many readers will be unfamiliar with the term ‘social networks’ or may assume that they are online social networks. Furthermore, throughout the background the social networks definition was poor.

Authors revision:

We appreciate your suggestion and have included a definition for lay social networks in the background, page 3, line 11-12 as follows: Informal social networks are an individual’s personal ties outside the formal medical system.(1)

4. Page 3, line 27-53: I am a little confused as to why the example provided relates only to a high-income country (UK), when the focus is on LMICs?

Authors revision:

Thank you for pointing this out. We have now included references from low- and middle-income countries and modified the previous text as shown below.

Original: For instance, a study examining the healthcare utilisation behaviours of Gypsies and travellers in the UK found connections between poor awareness and underutilisation of healthcare facilities and over-reliance on close social networks for health advice and support.(9)

Revised: Lay consultants contribute to health-seeking decisions, and their roles are particularly important in contexts where healthcare access is poor.(5) Some studies have found that lay consultants promote positive health-seeking behaviours, including encouraging prompt care-seeking and providing referrals to care providers.(6, 7). However, other studies have found that they discourage healthcare-seeking through rumours, or contribute to health-seeking delay due to time used in the consultation.(8, 9). (Page 3, 23-28)

5. There is little emphasis on specific networks in LMICs.

Authors revision:

Thank you for your suggestion. We have reworked the paragraph about networks in informal settlements of LMICs to highlight the specific issues as mentioned in the response to the overall comment on page 2-4 above.

Methods

1. Page 5, line 17: One of your research questions is 'who provides lay consultation', so why have you outlined in the eligibility criteria that social network members had to be related to the individual, such as family or friends, or community health workers? Surely this will narrow your findings?

Authors revision:

We acknowledge that there was an error in the inclusion criteria and have now modified it as shown below:

Original: The social network members had to be laypeople related to an individual, such as family members or friends.

Revised: The network members had to be lay people, e.g. family and friends. (Page 5, line 5-6)

2. Page 5, line 26: why were studies that looked at other health-related purposes besides symptoms or illness management included?

- We believe the reviewer means 'why wereexcluded'. Therefore, we have modified the eligibility criteria on as follows:

Previous: Studies were excluded if they were about: (a) lay consultation for other health-related purposes besides symptoms or illness management, such as disease prevention and pregnancy care; (b) long-term chronic conditions such as HIV or diabetes, where there was no emerging symptom or illness that required care different from the ways the individuals usually managed their health;

Revised: We excluded (a) studies on prevention and general health behaviours, as these are different from symptom and illness situations, which implies the perceived presence of abnormal health condition and need for care.(23) (b) Studies where the focus was on the management of a long term condition such as HIV or diabetes which had become part of the routine daily life for the individual and there were no new symptoms or illness experiences. (Page 5, line 10 -14)

- We have also clarified that the main focus of lay consultation is on symptoms and illness experiences in response to the reviewer's overall comment on page 1 and 2 above.

3. Page 7, line 11: patient and public involvement declaration should be moved to the end.

The information has now been moved to page 19, line 17-18.

Results

1. Page 12: You state that the themes were based on the research questions. Where is the rationale to support these research questions?

We have reworked the background to highlight the rationale for the study as shown from Page 3, line 29 to page 4, line 25:

This review considers the use and impacts of lay consultation in informal urban settlements of Low-and-Middle-Income Countries (LMICs). We acknowledge that "Informal urban settlements" is used interchangeably with "slums" in the literature, but we adopt the former as the latter may be defamatory.(10) Informal urban settlements are make-shift low resource settings in cities of LMICs, which houses more than 60% of the urban population.(11) These settlements are densely populated and lack basic social and physical amenities (including clean water, proper housing, hygienic

environment, security), contributing to a high burden of diseases.(12) Previous studies have found difficulties in accessing comprehensive formal medical care in informal urban settlements,(12, 13) and many rely on informal social support to cope with illnesses.(15) Understanding the use and roles of informal social networks during illness in such contexts is therefore important.

Lay consultation may be different in informal settlements for four major reasons. Firstly, informal social networks are the main sources of support during illness.(14) Secondly, while the demands are high, access to lay consultants can be challenging as the informal networks are limited and transitory.(15, 16) Thirdly, creation and sustenance of community social capitals (actual or potential resources obtainable from social networks)(17) are difficult, as informal settlers tend to be detached from non-kin community networks, and are often unable to provide continuous support to others to protect their mental and physical health.(18) Fourthly, informal settlers tend to lack access to weak ties, which are the social connections beyond an individual's immediate environment that can facilitate access to wider resources.(19) Thus, given these distinct challenges, it is important to understand the access to, use and consequences of lay consultation in the settlements.

There is an increasing call to engage social networks in facilitating health interventions in informal settlements of LMICs,(20) but this is hinged on understanding how the networks work. This review synthesises available evidence on the use and influence of lay consultants in the settlements to help policy makers and providers draw on their benefits to maximise value from healthcare and mitigate their negative effects. We aimed to answer three questions: (a) which informal social networks provide lay consultation? (b) what is the content of the lay consultation? (c) what are the consequences of lay consultation on health-seeking behaviours?

2. Page 12, line 5: For the third theme 'what are the consequences of lay consultation on health-seeking behaviors' – limiting the search to only symptoms and illness experience narrows your findings significantly.

We have clarified that lay consultation is specifically about illness and symptom experiences as mentioned in response to the reviewer's overall comment on page 1 and 2 above.

3. Why have you excluded other health related purposes? surely you have excluded other important findings relating to other health-related outcomes and health behaviors, in particular. For example, in LMICs, they have increased prevalence of certain health behaviors, i.e. smoking, so why were they excluded?

We agree that there are behaviours that have negative consequences for health that are common in LMIC such as smoking. We are aware that such behaviours are influenced by the social context in which people live. However, this review is about symptoms or illness experience.

Please see where we clarified that lay consultation is specifically about illness and symptom experiences in response to the reviewer's overall comment on page 1 and 2 above.

4. Page 13, line 12: this paragraph is missing citations.

The paragraph (now on page 13, line 11-18) contains 7 citations (please see below).

These refer to persons connected to an individual through friendship ties, engagement in similar activities or physical proximity. We found three groups under this category: friends,(34, 35) neighbours,(36-39) and colleagues.(30,35) Neighbours in shared structures were sometimes consulted or volunteered health advice when they noticed unusual symptoms in others.(38) While colleagues were common sources of advice, employers discouraged discussing health problems in the workplace.(30) As informal settlers were mostly casual workers, extra caution was employed in

seeking health advice from others to avoid wasting time or showing signs of poor health to their employers.(30)

The number of citations corroborates the frequency shown on the concept matrix on table 3 (page 12).

Discussion

1. Page 17, line 29: How did you assess if other formal forms of treatment/ health-care were also provided in conjunction with lay consultations?

One of our research questions was about the consequences of lay consultation on healthseeking behaviours (page 4, line 25). Thus, we extracted the relevant information from the included papers. Information on the positive and negative consequences is displayed on the concept matrix (table 3) on page 12.

2. Page 17, line 30-32: it is unsurprising that there was little by way of lay consultation as you state that you termed social networks to be related individuals in your eligibility criteria.

Our error on page 5, line 5-6 (see comment number 1 under the 'methods' section above) – meant the text was misleading. Social networks include people related to and not related to an individual.

3. Page 19, line 12: You introduce new terms in this paragraph but you have not explained them, i.e. 'weak ties' have not been defined.

We have now included a definition of weak ties in the background on page 4, 15-16: "social connections beyond one's immediate environment that can facilitate access to wider resources"

3. Page 12, line 14: please define 'norms of reciprocity'

The paragraph containing the phrase has been edited based on other reviewer's comment, and the phrase is no longer in the manuscript.

VERSION 2 – REVIEW

REVIEWER	Montgomery, Shannon Florida State University
REVIEW RETURNED	08-Sep-2021

GENERAL COMMENTS	I commend the authors on their careful and considerate revision of the manuscript. All queries were addressed.
--

REVIEWER	Rueger, Jasmina Wageningen University, Business Management & Organisation
REVIEW RETURNED	20-Sep-2021

GENERAL COMMENTS	Dear Authors, thank you for considering my comments when submitting a revised version of your manuscript. Firstly, I am not convinced that there is an added benefit in changing the title from an 'about' to an 'in', especially when the other reviewer requested to add something about 'health' or 'illness').
---

	Adding relevant theory and concepts established in the literature, such as that of the different types of social support (point 1), as well as adding relevant theory and updating the references to include more relevant and newer studies (point 8) has improved the overall quality of the manuscript. The sections elaborating on the exclusion criteria used for this study are still not as detailed as they could be, but I appreciate the efforts you have made to clarify them. Especially the part about informal healers, who may play an important role in LMICs, is much clearer. With regards to the decision to remove the part on digital communication, I believe this was a potentially interesting contribution, especially in the setting of the LMICs where access to 'traditional' healthcare providers is less extensive. However, since your manuscript does not include much detail about this, I agree that it should either be removed or left in a much shorter, less speculative form - as a suggestion for future research, for example. Lastly, my other comments regarding a lack of explanation of concepts such as 'weak ties' and 'lay consultation' have been addressed sufficiently. Thank you.
--	--